# Allergy Associated Myocardial Infarction: A Comprehensive Report of Clinical Presentation, Diagnosis and Management of Kounis Syndrome

**DOI:** 10.3390/vaccines10010038

**Published:** 2021-12-29

**Authors:** Anastasios Roumeliotis, Periklis Davlouros, Maria Anastasopoulou, Grigorios Tsigkas, Ioanna Koniari, Virginia Mplani, Georgios Hahalis, Nicholas G. Kounis

**Affiliations:** 1Department of Medicine, Mount Auburn Hospital, Harvard Medical School, Cambridge, MA 02138, USA; tasosdroumeliotis@gmail.com; 2Division of Cardiology, Department of Internal Medicine, Medical School, University of Patras, 26500 Patras, Greece; pdav@otenet.gr (P.D.); anastmaria89@hotmail.com (M.A.); gregtsig@gmail.com (G.T.); hahalisg@yahoo.com (G.H.); 3Manchester Heart Institute, Manchester University Foundation Trust, Manchester M23 9LT, UK; iokoniari@yahoo.gr; 4Intensive Care Unit, Medical School, University of Patras, 26500 Patras, Greece; virginiamplani@yahoo.gr

**Keywords:** acute coronary syndrome, allergy, anaphylaxis, Kounissyndrome

## Abstract

Kounis syndrome (KS) has been defined as acute coronary syndrome (ACS) in the context of a hypersensitivity reaction. Patients may present with normal coronary arteries (Type I), established coronary artery disease (Type II) or in-stent thrombosis and restenosis (Type III). We searched PubMed until 1 January 2020 for KS case reports. Patients with age <18 years, non-coronary vascular manifestations or without an established diagnosis were excluded. Information regarding patient demographics, medical history, presentation, allergic reaction trigger, angiography, laboratory values and management were extracted from every report. The data were pulled in a combined dataset. From 288 patients with KS, 57.6% had Type I, 24.7% Type II and 6.6% Type III, while 11.1% could not be classified. The mean age was 54.1 years and 70.6% were male. Most presented with a combination of cardiac and allergic symptoms, with medication being the most common trigger. Electrocardiographically, 75.1% had ST segment elevation with only 3.3% demonstrating no abnormalities. Coronary imaging was available in 84.8% of the patients, showing occlusive lesions (32.5%), vascular spasm (16.2%) or normal coronary arteries (51.3%). Revascularization was pursued in 29.4% of the cases. In conclusion, allergic reactions may be complicated by ACS. KS should be considered in the differential diagnosis of myocardial infarction with non-obstructive coronary arteries.

## 1. Introduction

It is well known that the cardiovascular system is heavily involved in allergic reactions with potential symptoms including hypotension, arrhythmias and ventricular dysfunction [1,2]. One not easily recognized, but possibly dangerous cardiac manifestation, is the development of acute coronary syndrome, also known as allergic angina [3]. This was initially described by Kounis and Zavras in 1991and was originally attributed to the effect of histamine release in the coronaries [4]. Since then, Kounis syndrome (KS), as commonly referenced in literature, has been associated with three distinct pathophysiologic processes stemming from mast cell activation and degranulation leading to the release of potent inflammatory mediators [5,6]. Type I refers to a coronary spasm in individuals with normal or near normal coronary vessels, type II to a coronary spasm or plaque rupture in patients with pre-existing coronary artery disease (CAD) and type III to stent thrombosis (subtype IIIa) or stent restenosis (subtype IIIb) following an allergic reaction [7]. Recent reports have associated KS with a significant prevalence (1.1%) amongst patients hospitalized following hypersensitivity reactions and a high subsequent mortality rate [8]. Despite this, it remains an under-diagnosed clinical entity in everyday practice [9]. In this review of previously published case reports, we aim to delineate the clinical and pathophysiologic characteristics of KS that render it unique and discuss optimal medical management for these patients.

## 2. Materials and Methods

We searched PubMed until 1 January 2020 for case reports and case series of patients presenting with KS. Using “Kounis [TIAB]” as our search parameters, we identified 517 unique publications that we subsequently reviewed. After excluding those without case reports, not carrying a definite KS diagnosis, including patients < 18 years old, or not being in English, we found a total of 247 papers presenting 276 patients. We additionally included another 12 case reports not originally diagnosed as KS but identified as such by letters to the editor of the respective journals. Overall, we were able to detect 288 cases of KS in prior literature (Figure 1, Appendix A).

After carefully reviewing all individual reports, we created a patient level database that was used for the present analysis. Multiple variables were collected for every case. The syndrome type was either specified by the original report or defined according to KS classification if enough information was available. Age, gender, pre-existing medical conditions and allergic reaction trigger were extracted for all of the patients. Regarding the clinical presentation of the syndrome, all initial symptoms were registered in our database and categorized into cardiac and allergic. Electrocardiographic findings on presentation were also available. We captured initial management (interventional vs. conservative) as well as medical therapy on discharge. In cases that coronary angiography or other coronary imaging studies were pursued, we compiled data on the coronary anatomy, pathophysiologic mechanism identified (clean coronaries vs. atherosclerosis vs. spasm) and revascularization strategy, if applicable. The initial and follow-up ejection fractions were included when available. If echocardiographic results were described but no specific percentage was mentioned, this was manually calculated by the authors. Finally, we assembled all of the accessible information on cardiac, allergic and inflammatory biomarkers. A detailed list of all collected parameters is summarized in Table 1.

The collected findings are reported as the mean or percentage for continuous and categorical variables, respectively. Only the patients with available data were included in the calculation of every parameter. In terms of biomarkers, the results were reported as elevated or not elevated rather than exact numbers to account for normal range variability between different labs. All descriptive statistics were analyzed using IBM SPSS Statistics.

## 3. Results

Of the 288 patients with KS that were analyzed, 71.2% were male and 28.8% were female and the average age was 54.1 years. The most common type was KS type I, while KS type III was the least frequent. Past medical history of a prior allergic reaction was documented in 31.4% and prior CAD in 21.9% of cases. Traditional cardiovascular risk factors including hypertension, diabetes mellitus, smoking and dyslipidemia were only present in a small fraction of patients (Table 2). With regards to the initial clinical presentation, a combination of ischemia and hypersensitivity symptoms was reported in most cases. Chest pain was the most common complaint, but skin rashes, itching or swelling were also present in multiple reports. Interestingly, hypotension was found in 50% of patients. The frequency of presenting symptomatology is summarized in Figure 2. An allergic reaction trigger was recognized in 93.7% of cases. The most prevalent culprit was adverse reaction to medication, with antibiotics being the leading cause. Other triggers included insect bites, food consumption, systemic disease manifestation, environmental exposure and contrast agent administration (Table 3). A detailed list of all of the Kounis syndrome triggers can be found in Table 4.

As part of the standard acute coronary syndrome work-up, electrocardiography and cardiac biomarkers were available for almost all of the patients. The most common finding was ST elevation while only 3.3% of patients had a normal EKG and troponin elevation was seen in 77.5% of patients. Despite this impressive clinical presentation, 50.6% of the subsequently performed catheterization procedures failed to reveal any changes in the coronary arteries and hemodynamically significant lesions due to atheromatous or thrombotic lesions requiring revascularization were only found in one third of the cases. The most frequently involved vessels were the left anterior descending and the right coronary artery (Table 2).

Inflammatory cells and biomarkers that are frequently implicated in allergic reactions include tryptase, eosinophils and immunoglobulin E (IgE). Amongst cases with reported biomarker values, elevation of tryptase, eosinophils and IgE was found in 80.6%, 58.0% and 75.7% of cases, respectively.

Finally, the long-term management of patients with KS was variable. Antiplatelet therapy following the hospitalization was consistent with the results of the cardiac catheterization, with 41.2% getting discharged on aspirin and 35.1% on a P2Y_12_. With regards to the allergic reaction, multiple patients received steroids, epinephrine, calcium channel blockers and/or antihistamines but again no reliable pattern was identified.

## 4. Discussion

The main findings of the present review include: (1) allergic reactions may be complicated with acute coronary syndrome, (2) many patients with KS present in a young age and have no risk factors of CAD or history of prior allergic reaction, (3) even though most of the patients present with ST-elevation myocardial infarction, only a small fraction will require coronary revascularization, (4) many inflammatory biomarkers including tryptase, eosinophils and IgE have been found to be elevated and(5) no algorithm for the management of KS is currently available.

Allergic reactions have multiple manifestations, including skin and mucosal surface changes, gastrointestinal tract upset, respiratory distress as well as cardiovascular symptoms. Our review, as well as previous studies, demonstrate that hypersensitivity may be complicated by acute coronary syndrome and myocardial ischemia [10]. This finding comes in line with the growing body of evidence correlating inflammation with CAD and atherosclerosis, also known as the inflammatory hypothesis [11]. Following percutaneous coronary intervention persistent high residual inflammatory risk, defined as serial elevated high sensitivity C-reactive protein measurements, has been associated with a worse prognosis [12,13]. In addition, reducing inflammation by using canakinumab, a monoclonal antibody against interleukin-1β, improved long-term cardiovascular outcomes in the Canakinumab Anti-inflammatory Thrombosis Outcome Study (CANTOS) trial [14]. Interestingly, interleukin-1β has been shown to be implicated in multiple allergic disorders [15].

Inflammatory cells, including eosinophils, macrophages and mast cells, play a significant role in hypersensitivity reactions. They initiate the inflammatory cascade by releasing mediators, including leukotrienes, thromboxane, IgE, tryptase, histamine and cytokines, that act on multiple targets located on various organ systems [16]. At the level of the coronaries, histamine can cause vasoconstriction that appears to be the main pathophysiologic mechanism in KS type I [17]. Other mediators accelerate the atherosclerotic process and facilitate plaque rupture that may lead to KS type II [18]. Finally, this process may promote impaired endothelial healing after drug eluting stent implantation which, combined with platelet activation by leukotrienes, thromboxane and proteolytic enzymes during an acute allergic reaction, can result in stent thrombosis observed as part of KS type III [19]. In this report, eosinophils, tryptase and IgE were found to be elevated in a significant number of patients supporting the diagnosis of KS. Even though assays to measure all inflammatory markers are not readily available in most hospitals, we believe that a complete blood count with manual differential, to look for eosinophils, is readily available and should be routinely ordered in patients with a clinical history suspicious of KS.

A wide variety of KS triggers has been previously described [10]. In our review, medications and, more precisely, antibiotics, were shown to be the most common etiology but other exposures, including contrast media, insect bites, and food, were also prevalent (Table 3). Interestingly, flairs of systemic inflammatory diseases such as asthma, systemic mastocytosis or recurrent urticaria were the initial trigger in 2.4% of cases. This finding lends supports to prior reports suggesting that hypersensitivity disorders may constitute a risk factor for CAD [20,21].

Clinical presentation and management of patients with KS also appears to be of a particular interest. Chest pain was the most common presenting symptom and most patients also exhibited allergic symptoms such as arash, swelling and itching. Despite this, cases without cardiac symptomatology found to have ischemia either by EKG changes or biomarker elevation have been reported [22]. On the other hand, there have been reports of patients presenting with ACS as the sole manifestation of a hypersensitivity reaction [23]. This wide variance in clinical presentation as well as the absence of cardiac risk factors may be the reason why KS remains an underdiagnosed entity while evidence from the national inpatient sample showed that 1.1% of patients admitted with allergic, hypersensitivity or anaphylactic reactions were concomitantly diagnosed with an ACS [8].

Regarding the management of KS, we observed wide variability between cases, likely stemming from the absence of predetermined algorithms. Optimal treatment consists of concomitant management of myocardial ischemia and the allergic reaction. This can be clinically challenging since medications treating one condition may exacerbate the other [24]. The use of intravenous fluids that is required in patients with distributive shock from anaphylaxis but can precipitate pulmonary edema in those with cardiogenic shock from ACS constitutes a prime example of this principle.

With regards to managing the allergic component, we found that corticosteroids were the most used medication on admission, and given their limited cardiac side effects, their use in KS appears to be safe. Antihistamines that were the second most utilized class are also safe in patients with ACS but must be used with caution since rapid intravenous administration may lower blood pressure [25]. Finally, epinephrine, which is the gold standard for the treatment of life-threatening complications of anaphylaxis, was used in approximately one third of the cases we reviewed. Despite this, clinicians should be aware of the potential side effects of this medication, including coronary vasospasm and arrhythmias, that may deteriorate cardiac function. As a result, it should only be administered in a closely supervised setting and the intramuscular route should be preferred to reduce the potential side effects [26].

The management of ACS in the acute phase should follow the existing guideline recommendations primarily based on electrocardiographic criteria, since no specific recommendations are currently available [27,28]. In our review, we found that aspirin was prescribed to 41.2% of patients and dual antiplatelet therapy was only initiated in patients requiring stent placement. Beta blockers should be avoided given the possibility for worsening coronary vasospasm due to uncontested α-adrenergic activity [29]. In the Type I variant of KS, for which coronary vasospasm is the primary pathophysiologic mechanism for coronary hypoperfusion, systemic or intracoronary nitrates as well as calcium channel blockers, such as diltiazem and verapamil, were frequently utilized and appear to be an acceptable treatment options for such patients. Though, it is important to highlight the need for close hemodynamic monitoring since they may dangerously lower the blood pressure in the context of an allergic reaction. This is further supported by our finding of hypotension on admission of approximately half the included patients. These medications were also continued at discharge for a considerable number of patients but evidence on long-term outcomes and complications for patients presenting with KS is currently scarce even though cases of recurrent disease have previously been described [30]. Based on our findings, an algorithm for optimal management of patients with KS is proposed in Figure 3.

Finally, patients with KS type I presenting with ST-segment elevation myocardial infarction in the absence of obstructive coronary disease in KS more likely suggests resolved epicardial spasm and/or resolved or ongoing acute microvascular spasm/disturbance. Although the pathophysiology, course, treatment and prognosis require further investigation, we think that KS Type I meets the criteria for myocardial infarction in the absence of obstructive coronary artery disease (MINOCA) and should be incorporated in future guidelines. Such patients should also be further evaluated and treated according to the current MINOCA recommendations [31].

## 5. Limitations

Important limitations should be considered when interpreting our results. Stemming from previously published case reports, our findings are subject to publication bias. There was no uniformity in data presentation across different included reports, resulting in a significant number of missing variables in the created database. Coronary angiography, which is required for definite classification of KS type, was not performed in some cases. We included all cases of KS identified as such by the authors or letters to the editor of the journals. Despite this, given that KS represents an underdiagnosed entity, we may have missed other cases published in literature but not accordingly labeled.

## 6. Conclusions

In conclusion, hypersensitivity reactions regardless of the initial trigger may be complicated by ACS, and KS appears to be an underdiagnosed entity. Management should be tailored to the underlying pathophysiology according to the most recent guidelines for ACS management. KS type I should be considered in the differential diagnosis of MINOCA.

## Figures and Tables

**Figure 1 vaccines-10-00038-f001:**
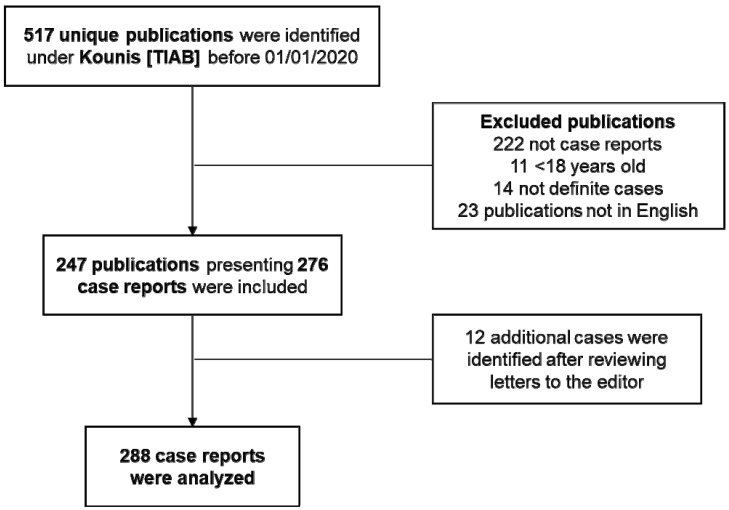
Flowchart showing the selection process for the included papers and exclusion criteria.

**Figure 2 vaccines-10-00038-f002:**
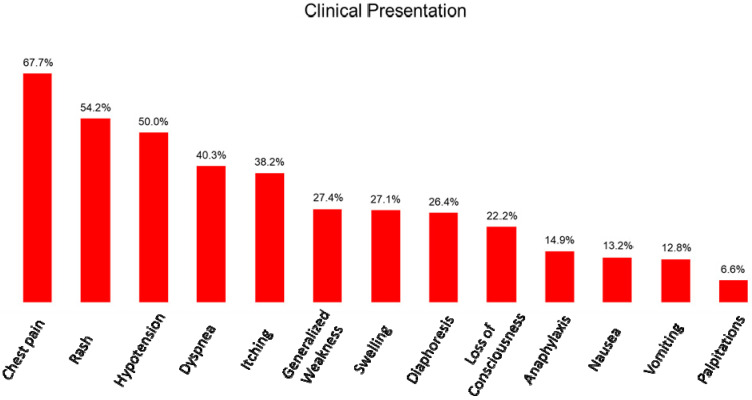
Clinical presentation of patients with Kounis syndrome according to symptom prevalence.

**Figure 3 vaccines-10-00038-f003:**
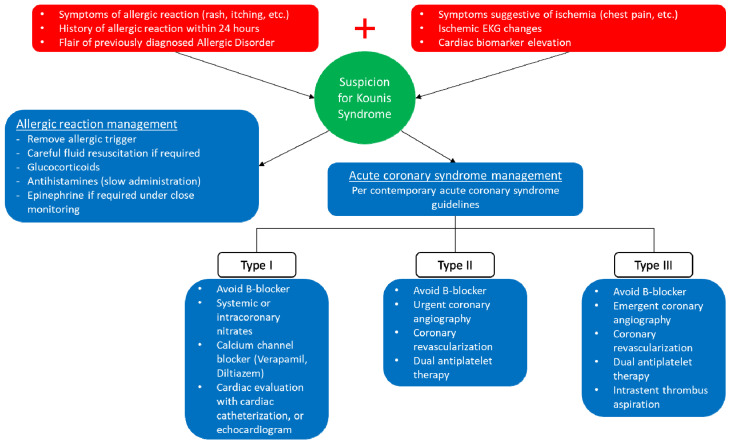
Suggested algorithm for Kounis syndrome management.

**Table 1 vaccines-10-00038-t001:** List of extracted parameters.

Kounis Syndrome Type	Past Medical History
**Patient Age**	Coronary artery disease
**Trigger of the allergic reaction**	Prior percutaneous coronary intervention
Antibiotics	Peripheral arterial disease
Anesthetic drugs	Diabetes mellitus type I
Cardiac drugs	Diabetes mellitus type II
Other drugs	Chronic kidney disease
Insect bite	Hypertension
Food	Smoking
Systemic disease	Dyslipidemia
**Symptomatology on presentation**	Family history of coronary artery disease
Dyspnea	Overweight
Itching	History of allergy
Rash	Other
Swelling	**Laboratory results**
Sweating	Troponin
Nausea	CPK
Vomiting	CK-MB
Hypotension	Eosinophils
Anaphylactic shock	IgE
Cardiac symptoms	Specific IgE
Squeezing pain	Tryptase
Loss of consciousness	**Medication at discharge**
General symptoms	Aspirin
Palpitations	P2Y12
**Electrocardiographic changes**	Heparin
ST elevation	Calcium channel blockers
Other acute ischemic changes	Nitrates
Bundle branch block	Antihistamine
Other ECG	**Catheterization results**
**Allergic reaction management**	Left anterior descending involvement
Epinephrine	Left circumflex involvement
Cortisone	Right coronary artery involvement
Antihistamine	Left main involvement
Nitrates	Atheromatosis/thrombosis
**Echocardiography**	Spasm
Previous Known EF	Clean coronary arteries
Present EF	**Acute coronary syndrome management**
	Thrombolysis
	Coronary revascularization

**Table 2 vaccines-10-00038-t002:** Baseline clinical characteristics of patient presenting with Kounis Syndrome.

Clinical Characteristics	Prevalence
Sex	
Male	71.20%
Female	28.80%
Kounis syndrome type	
I	57.60%
II	24.70%
III	6.60%
Unclassified	11.10%
Past medical history	
Coronary artery disease	21.90%
Vascular disease	4.50%
Diabetes mellitus	15.30%
Chronic kidney disease	1.70%
Hypertension	31.40%
Smoking	21.50%
Dyslipidemia	18.60%
Overweight	5.80%
Prior allergic reaction	31.40%
Troponin elevation	77.50%
Electrocardiographic findings	
ST segment elevation	76.20%
T waves inversion and/or ST segment depression	20.40%
Normal EKG	3.30%
Coronary artery involved	
Left main	2.50%
Left anterior descending	27.30%
Left circumflex	12.20%
Right coronary artery	26.50%
Catheterization findings	
No changes	50.60%
Coronary thrombosis/atheromatosis	33.10%
Coronary spasm	16.30%

**Table 3 vaccines-10-00038-t003:** Incidence of the triggers leading to allergic reaction complicated by Kounis Syndrome.

Triggers	Incidence
Unknown	6.3%
Medications	51.7%
- Antibiotics	44.3%
- Anesthetics	9.4%
- Cardiovascular	6.7%
- Other	39.6%
Insect bite	18.8%
- Bee	50.0%
- Wasp	29.6%
- Other	20.4%
Food	9.7%
- Seafood	42.9%
- Fruit	17.9%
- Other	39.2%
Systemic disease	2.4%
Environment	1.4%
Contrast	6.3%
Other	3.5%

**Table 4 vaccines-10-00038-t004:** Detailed list of all previously identified triggers.

MEDICATIONS	
**Antimicrobials**	**Other**
Amoxicillin/clavulanic acid	Ranitidine
Ampicillin/sulbactam	Dextran 40
Cefazolin	Viper antiserum
Penicillin	Gelofusine
Penicillin G	Succinylated gelatin
Amoxicillin	Recombinant human insulin
Cefotaxime	Triamcinolone
Cephalosporin	Hyoscine butylbromide
Ceftriaxone	Cervus and cucumis polypeptide
Cefuroxime	Dextromethorphan
Cefaclor	Progesterone
Cefuroxime axetil	Low molecular weight dextran
Ceftazidime	**FOOD**
Piperacillin/tazobactam	Blue crab
Cefditoren pivoxil	Gilthead
Moxifloxacin	Canned tuna fish
Ciprofloxacin	Cephalus fish
Levofloxacin	Tuna
Gemifloxacin	Tuna sandwich
Metronidazole	Shellfish
Fluconazole	Prawns
Clarithromycin	Anchovies
Brivudine	Fish
Telithromycin	Fish eggs
Trimethoprim/sulfamethoxazole	Undercooked fish
Clindamycin	Blue skinned fish
Vancomycin	Pleurotus ostreatus
**Cardiac Medications**	Mushrooms
Amiodarone	Rice pudding
Lidocaine hydrochloride	Lentil
Aspirin	Kiwi
Clopidogrel	Mango
ACE I	Pineapple
Epinephrine	Pea salad
**Chemotherapeutic Agents**	Salad with mustard
Cisplatin	Milk
Capecitabine	Wheat
Rituximab	Scallion
Oxaliplatin	Meat
Carboplatin	**SYSTEMIC DISEASE**
Daratumumab	Systemic macrocytosis
Paclitaxel	Asthma exacerbations
Cyclophosphamide	Urticaria flair
5-fluorodeoxyuridylate	Chronic recurrent urticaria
**Anesthetics**	Idiopathic autoimmune urticaria
Bupivacaine	**INSECT BITE**
Cisatracurium	Bee sting
Succinylcholine	Bumblebee sting
Rocuronium	Honeybee sting
Mepivacaine	Wasp sting
Atracurium	Hornet sting
Propofol	Viper bite
Rocuronium-sugammadex	Adder bite
**Analgesics**	Snake venom
Ibuprofen	Cobra bite
Acetaminophen	Scorpion
Celecoxib	Spider bite
Metamizole	Warble fly bite
Diclofenac sodium	Fire ant bite
Diclofenac potassium	Fly carvaria
Ketoprofen	Larvae
Paracetamol	**ENVIRONMENTAL FACTORS**
Naproxen sodium	Dust
Acemetacin	Cold sea water contact
Propyphenazone	Kinds of metal
Flavoxate	Physical exercise
Tramadol	Post viral
Meperidine	Fishbone injury
Remifentanil	Plant
Morphine	Chinese herbs
Ketorolac	Pesticide sprays
**Cannabinoids**	Tree’s fluff
Marijuana	**OTHER TRIGGERS**
Bonsai type synthetic drugs	Leech therapy
**Proton Pump Inhibitors**	Latex
Pantoprazole	Ruptured hepatic echinococcal cyst
Lansoprazole	Combination of animal protein
**Decongestants**	Hemodialysis apparatus
Pseudoephedrine	Food-dependent exercise-induced anaphylaxis
**Antipsychotics**	Unknown
Ziprasidone	
Midazolam	
**Contrast Media**	
Iopromide	
Echo contrast media	
Iodinated contrast media	
Radioiodine contrast media	
Gadolinium contrast media	
Gadoteric acid	

## Data Availability

Not applicable.

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
