# Peer review of "Allergy Associated Myocardial Infarction: A Comprehensive Report of Clinical Presentation, Diagnosis and Management of Kounis Syndrome"

_vaccines, 2021, doi:10.3390/vaccines10010038_

Round 1
Reviewer 1 Report
I have reviewed the manuscript termed "Allergy Associated Myocardial Infarction: A Comprehensive Report of Clinical Presentation, Diagnosis and Management of Kounis Syndrome" by Roumeliotis and colleagues. In this study the authors have searched the PubMed for reports that describe cases or case series of the Kounis syndrome, and from which generated a data base where several parameters are registered. I acknowledge that the last author also was the person first describing this syndrome.
This is a very well written article describing 288 cases with Kounis syndrome, and from this dechiffer: the triggering factors, the distribution between the different types, gender distribution etc. The importance of this study is that it will make the field aware of the disorder, otherwise there might be a risk for mis-classification of the Kounis syndrome for MINOCA or Takotsubo.
Even though it is well written, there some minor concerns that need to be addressed:
1) 57.6% of the patients were classified as type 1, but only 50.6% of the patients have undergone an angiogram. How do you know that the patients have no atheromatosis that could re-classify the patients to type II?
2) Smoking has been misspelled at line 98.
3) The wording "manuscript" is commonly used in the text. To most readers, a manuscript is a report that has not been peer-reviewed. Please change the wording to "previous reports", "studies" or similar.
Author Response
Reviewer 1
I have reviewed the manuscript termed "Allergy Associated Myocardial Infarction: A Comprehensive Report of Clinical Presentation, Diagnosis and Management of Kounis Syndrome" by Roumeliotis and colleagues. In this study the authors have searched the PubMed for reports that describe cases or case series of the Kounis syndrome, and from which generated a data base where several parameters are registered. I acknowledge that the last author also was the person first describing this syndrome.
This is a very well written article describing 288 cases with Kounis syndrome, and from this dechiffer: the triggering factors, the distribution between the different types, gender distribution etc. The importance of this study is that it will make the field aware of the disorder, otherwise there might be a risk for mis-classification of the Kounis syndrome for MINOCA or Takotsubo.
Even though it is well written, there some minor concerns that need to be addressed:
1) 57.6% of the patients were classified as type 1, but only 50.6% of the patients have undergone an angiogram. How do you know that the patients have no atheromatosis that could re-classify the patients to type II?
We thank the reviewer for this very important comment. It is accurate that not all patients had undergone coronary angiography to classify the type of Kounis syndrome since it was not deemed as clinically appropriate by the treatment team.In some of those cases non-invasive imaging with coronary CT angiography was available for the assessment of coronary anatomy. For all cases we registered the type of Kounis syndrome as classified by the authors of the case report/series. The limitations section has been edited appropriately.
“…resulting in a significant number of missing variables in the created database. Coronary angiography which is required for definite classification of KS type was not performed in some cases. We included all cases of KS identified as such by the authors…”
2) Smoking has been misspelled at line 98.
We thank the reviewer for this comment. Spelling has been corrected as appropriate.
3) The wording "manuscript" is commonly used in the text. To most readers, a manuscript is a report that has not been peer-reviewed. Please change the wording to "previous reports", "studies" or similar.
We appreciate this important remark by the reviewer. The wording “manuscript” has been changed as appropriate throughout the text.
=x/c/v
+X/C/V
+X/C/V
Reviewer 2 Report
Thank you for permitting me to review this manuscript
Line 52 please insert a number for (significant prevalence)
Line 204 The authors suggest Intramuscular injection of epinephrine , this is not a classical mode of administration , in anaphylaxis situations, the administration of epinephrine, if necessary is usually Intravenous, therefore I suggest to nuance this sentence and consider different situations in which IV administration is mandatory.
Table 3
arterial chemoembolisation per se is not a chemotherapeutic agent as is general anesthesia , or dental infilttration anesthesia , which is indeed local anesthetics
please correct iodinated contrast
succinylcholine which is a muscle relaxant is not reported , however allergic reactions with this drug is high please explain or add
Author Response
Reviewer 2
Thank you for permitting me to review this manuscript
Line 52 please insert a number for (significant prevalence)
We thank the reviewer for this comment. According to the analysis form the United States nationwide database from 2007-2014 almost 1.1% of patient admitted withallergy/hypersensitivity/anaphylactic reactions had Kounis syndrome. The introduction has been edited appropriately:
“…have associated KS with a significant prevalence (1.1%) amongst patients hospitalized following hypersensitivity…”
Line 204 The authors suggest Intramuscular injection of epinephrine , this is not a classical mode of administration , in anaphylaxis situations, the administration of epinephrine, if necessary is usually Intravenous, therefore I suggest to nuance this sentence and consider different situations in which IV administration is mandatory.
We thank the reviewer for this important comment. It is accurate that epinephrine in patients with anaphylactic shock in the ER settingis classically administered via the intravenous route. As we mention in the Discussion section of our review patients with Kounis syndrome are unique since they tend to exhibit a combination of distributive and cardiogenic symptomatology rather than a pure distributive shock. As a result, potential cardiac side effects of epinephrine including coronary vasospasm and arrhythmias may be detrimental. Intramuscular epinephrine administration which is the preferred route in the non-hospital setting has been shown to have a better side effect profile and therefore should be preferred in this clinical scenario.
Table 3
Arterial chemoembolisation per se is not a chemotherapeutic agent as is general anesthesia , or dental infilttration anesthesia , which is indeed local anesthetics
Please correct iodinated contrast
Succinylcholine which is a muscle relaxant is not reported, however allergic reactions with this drug is high please explain or add
We thank the reviewer for his important remarks. The table has been edited appropriately:
- Chemoembolization, general anesthesia, and dental infiltration anesthesia have been removed.
- Iodinated contrast has been edited appropriately.
- Succinylcholine has been added to the table under Anesthetics.